# WIC Staff Views and Perceptions on the Relationship between Food Insecurity and Perinatal Depression

**DOI:** 10.3390/healthcare11010068

**Published:** 2022-12-26

**Authors:** Karen M. Tabb, Shannon D. Simonovich, Jana D. Wozniak, Jennifer M. Barton, Wan-Jung Hsieh, Claire Klement, Mary Ellen Ostrowski, Noreen Lakhani, Brandon S. Meline, Hsiang Huang

**Affiliations:** 1School of Social Work, University of Illinois at Urbana-Champaign, Urbana, IL 61801, USA; 2Beckman Institute for the Advancement of Technology, University of Illinois at Urbana-Champaign, Urbana, IL 61801, USA; 3School of Nursing, College of Science & Health, DePaul University, Chicago, IL 60614, USA; 4Department of Psychiatry, Cambridge Health Alliance, Harvard Medical School, Cambridge, MA 02139, USA; 5Family Resiliency Center, University of Illinois at Urbana-Champaign, Urbana, IL 61801, USA; 6Department of Social Work, National Taiwan University, Taipei 10617, Taiwan; 7Maternal and Child Health Division, Champaign-Urbana Public Health District, Champaign, IL 61820, USA

**Keywords:** food insecurity, WIC, low-income, depression, nonmetropolitan

## Abstract

Food insecurity and perinatal depression are significant public health concerns for perinatal services, however descriptive research examining their association is limited. The purpose of this study was to examine the views and perspectives of staff from the Special Supplemental Nutrition Program for Women, Infants and Children (WIC) program on the relationship between food insecurity and perinatal depression among their WIC clients. Four, semi-structured focus groups with WIC staff (*n* = 24) were conducted across four diverse nonmetropolitan public health districts in Midwestern counties in the United States. WIC staff included social workers, nurses, nutritionists and ancillary staff. All interviews were audio-recorded, transcribed, and verified, and data were organized using NVivo 11.4.2. Thematic networking analysis was employed as the qualitative analysis to identify organizing themes. Three themes emerged including (1) depression experienced by clients; (2) food insecurity experienced by clients; and (3) barriers preventing clients from accessing services for themselves and their children. Research on food insecurity and perinatal depression is sparse, with fewer studies having included health staff of low-income women. Our findings suggest that the association between food insecurity and mental health needs among WIC clients is a significant public health issue to which policy change and interventions are required.

## 1. Introduction

Food insecurity among women in the perinatal (i.e., pregnant or postpartum) period is a significant public health concern that is associated with considerable physical and mental health concerns. The United States Department of Agriculture (USDA) conceptualizes food insecurity as not having consistent access to the amount of food necessary to lead an active or healthy life for all family members [1]. One concern related to the experience of food insecurity in pregnant women is that of perinatal depression. Research findings demonstrate that pregnant, food insecure women are 2.5 times more likely to experience depression as compared to their food secure counterparts [2,3]. Data also supports that the experience of depression can increase the risk of food insecurity. Noonan and colleagues [4] found that the risk of food insecurity increases by 50% to 80% when a woman is moderately to severely depressed. Taken together, there may be a bidirectional relationship between food insecurity and depression.

Socioeconomic status is also an influential factor when examining the relationship between reliable access to adequate food and depressive symptoms. For example, rates of food insecurity and postpartum depression are both higher in low-income women [5] who are more likely to live in “stressed” environments. Living in an environment that is considered “stressed” (i.e., an environment with low access to material goods and high residential mobility) is associated with greater incidence of depression as compared to less stressed environments [6]. “Stressed” environments—a term which is often a proxy measure used for conceptualizing the experience of poverty—can also often be food deserts, meaning they have significantly limited access to healthy, affordable food. Thus, individuals living in a low-income context independently experience greater food insecurity and depressive symptoms than those who do not. 

Perinatal depression and food insecurity each separately contribute to declines in maternal and child physical and psychological health. Perinatal depression is a challenge because it not only affects women’s health but also the health of infants, including low birthweight and early cessation of breastfeeding, and subsequent long term cognitive and behavioral concerns in children (see Slomian, et al. [7] for review). Children are more likely to struggle with behavioral and cognitive issues when their households experience food insecurity, especially if the mother is experiencing depression [8]. Additionally, children from households who experience food insecurity are at a greater risk for developing cardiovascular complications, diabetes, and hyperlipidemia later in life [9]. 

Given the significant impact that food insecurity and depression have on low-income women and their families, it is imperative to identify markers for women may be experiencing either or both. Healthcare staff who meet with these women across the perinatal period play a pivotal role in identifying those who may be at risk or currently struggling. Providers and staff are often the first point of contact for women experiencing perinatal depression symptoms, as their clinics are often tasked with screening for mood symptoms and psychosocial stability, as well as the provision of related resources [10]. 

One particularly helpful resource available to women in the perinatal period is the Special Supplemental Nutrition Program for Women, Infants, and Children (WIC). The WIC program provides healthy food, nutritional counseling, and healthcare services to low-income women and children across the United States (U.S.). Increased length of participation in WIC has been linked to improved food security status for an ethnically diverse sample of mothers and children [11,12] and longitudinal evidence revealed that each additional WIC clinic visit significantly decreased household food insecurity [11]. Thus, participation in WIC may be a promising avenue toward understanding the connection between improvements in food insecurity, depressive symptoms, and overall functioning during the perinatal period. However, there is limited qualitative evidence describing how WIC staff view the experience of food insecurity and depressive symptoms among their clients, and to address this gap. Thus, the purpose of the current study was to describe how WIC staff perceive, identify and address the needs of low-income women, and how those needs relate to food insecurity and depression.

## 2. Methods

### 2.1. Setting

The setting for this study is three WIC programs housed within public health clinics in nonmetropolitan and rural counties in a Midwestern state in the U.S. Each of these three counties has a population density less than 150,000, including rural and micro-urban areas. Collectively, these nonmetropolitan counties are home to more than 20,000 migrants, immigrants, and refugees, with growing French-Congolese and Central American populations. These clinics are responsible for administering the WIC program to nutritionally at-risk low-income childbearing women and their children up to age 5 residing in the county. Each family’s food insecurity status is one of several factors discussed between WIC staff and program participants during their visits. 

### 2.2. Data Collection and Data Analysis

During January 2018, we conducted four focus groups—one focus group at each of two clinic locations and two focus groups at the third location—with a total of 24 participants working as WIC-providing public health clinic staff members. Two trained facilitators (KT, SDS) concurrently led each of the four groups. Participants provided informed consent and completed a demographic questionnaire prior to the start of the interview. The focus group interviews followed a semi-structured interview guide with a set of questions related to assessment of food insecurity in WIC families with particular interest in culturally diverse populations and concealment of food insecurity. In addition to the interview guide, probing questions clarified responses of study participants to interview questions. Focus groups interviews lasted until saturation of emerging themes were met.

Focus group interviews lasted 45–55 min in length, were audio recorded, and were transcribed verbatim and verified (MEO, CK). Data entry of demographic questionnaire data was completed (SDS) and was then analyzed utilizing SPSS 24. The focus group interviews were analyzed via thematic analysis utilizing Nvivo 11.4.2 software (QSR International Pty Ltd. (2020) NVivo (released in March 2020), https://www.qsrinternational.com/nvivo-qualitative-data-analysis-software/home). The analysis was conducted by five of the authors (K.T., S.D.S., M.E.O., C.K., N.L.) who read each of the four transcriptions and identified codes. Following identification of codes, the raters met to discuss differences and similarities amongst codes to ensure consistency. Codes were refined and discussed until consensus amongst raters was reached. For the purposes of this paper, codes related to WIC staffs’ adaptations in assessing and addressing food insecurity were synthesized into three key themes expressing the collective views and experiences of our study participants.

### 2.3. Ethical Approval

All data collection processes, procedures and related paperwork received proper approval from the DePaul University Institutional Review Board, University of Illinois Institutional Review Board and local public health research integrity committees prior to initiation of the research study. 

### 2.4. WIC Clinic Staff Participants

A convenience sample of 24 WIC public health clinic staff were recruited through announcements in staff meetings, emails sent to their professional accounts and word of mouth. Participants included staff with backgrounds in diverse fields including: four public health nurses (16.70%), five social workers (20.80%), five dietitians/nutritionists (20.80%), four clerical staff (16.70%) and six “other” (25.00%). Study participants had an average of nine years in public health practice and ranged from 0.90 to 26.75 (*SD* = 7.70). Study participants reported, on average, having served in their current position for 7 years, range 0.20 to 24.00 (*SD* = 6.00). Participants reported having completed a Bachelor’s degree (41.70%), a Master’s degree (25.00%), an Associate’s degree (16.70%), some college (12.50%), or a high school diploma (4.20%). The majority of participants spoke English only; however, 25% of participants in the study sample spoke Spanish in addition to English, and one participant was fluent in Mandarin. Amongst WIC staff members who self-reported their race 70.80% identified as White, 16.70% as Black or African American, and 8.30% as Asian. In addition, 12.50% of the study sample identified as Latinx.

## 3. Results

We found three organizing themes that were related to WIC staffs’ perceptions of depressed mood and related food insecurity among participating families. As illustrated in see Figure 1 the three organizing themes relate to a series of key subthemes. 

### 3.1. Depression Experienced by Perinatal WIC Clients

The first theme, *Depression Experienced by Perinatal WIC Clients*, illustrated the negative affect of maternal depression or depressive symptoms on their ability to care for themselves and their families. Subthemes such as social isolation, cycle of worries and concerns, and lack of mental health resources were identified as factors that exacerbate and perpetuate symptoms. When asked how mothers describe their depressive symptoms, one participant described the experience as:


*“It’s like a cloud … They don’t see clear[ly] what’s going on … they need to [care for their family] … but they don’t have the energy. It’s not just affecting them; it’s affecting the whole family.”*


In addition to the experience of depressive symptoms in the perinatal period, one WIC staff described their views on the relationship between maternal depression and food insecurity as an all-encompassing cycle of worries: 


*It goes to all those … daily stressors, every single day. So if … you are constantly worried about not having enough money, not having enough food, not having transportation. All of those things. And that feeds into depression and anxiety. It takes every single thing you have to get out of bed in the morning and get here. And maybe some days it’s not an option to get [to WIC] because you can’t.*


From the perspective of the WIC staff, depression is not limited to the experience of depressed mood or anhedonia for these individuals; rather, the experience of lack of resources, daily stressors, and food insecurity during the perinatal period can exacerbate the mother’s emotional state. WIC staff also highlighted how depressive symptoms such as low energy or emotional withdrawal can uniquely affect a mother’s ability to activate in the face of food insecurity or psychosocial stressors. When describing how WIC clients recognize that they are food insecure while also experiencing perinatal depression, one participant shared: 


*It’s a hopelessness isolation ... I feel like the moms sense they don’t have food but [they don’t] have that urgency ... mentally. Mom can [go] through severe ... postpartum depression ... with good family support [but] they are feeling very isolated. They could have 10,000 people around [but they] encapsulate themselves, [so they] cannot really sense ... what’s surrounding them. It’s a protection ... mechanism. I see that a lot ... A lot of women do not speak [the] language, [and are] without transportation. They have to depend on [their] husband [to] drive them but [he] can be stuck in school ... I have [a mom] that experience[s] quite severe [depression], and she doesn’t even feed her baby. She doesn’t even remember the baby ... I mean you think rationally she loves her baby. She knows what she needs to do, but that’s what I [am] working [on] with her ... In my mind, [I] think about it like a lonely island. She’s there and ... she cannot see [a] way out. And then ... you can have so many people around her but that loneliness is very real to her.*


Taken together, WIC staff identified perinatal depression as a significant concern in its own right, and that symptoms can both intensify a mother’s experience of food insecurity and be exacerbated by the many challenges this population faces. 

### 3.2. Food Insecurity in Perinatal WIC Clients

The second theme, *Food Insecurity in Perinatal WIC Clients,* can be defined as challenges or barriers that WIC clients face in providing nutritious food for themselves and their families. In addition to the impact of depressive symptoms on food insecurity, financial concerns and the experience of stigma related to expressing the need for food were identified as subthemes. WIC staff primarily highlighted the disparity between health professionals’ expectations of dietary recommendations for food consumption in a given day per the USDA (2020) and the reality of the WIC clients’ experiences. One participant stated: 


*“In order to eat healthy like you’re supposed to, you’re supposed to eat six times a day … three good meals, three good snacks. These women don’t have that. They are lucky if they get one meal.”*


Participants noted that perinatal WIC clients often cannot access sufficient nutrition for themselves or their families. Beyond the cost of the food itself, participants identified additional barriers that impede access to food. For example, participants noted that women experiencing depression might also find it difficult to mobilize or motivate themselves to go to the store or to the WIC office, or to cook for their families when they do have food. 

A second barrier, lack of adequate transportation, was also identified by WIC staff. Participants noted that, some WIC clients have access to convenience stores within walking or bussing distance, yet these markets tend to lack the kinds of food needed for optimal nutrition (e.g., fruits, vegetables, whole grains). One participant described: 


*Even [if] they get coupons, they don’t have transportation [to] take them to the big supermarket to get the fresh produce. And that’s a hinder[ence] to access healthier food because they may have no transportation to get them to the big market where have more choice of fresh produce than if they go to little convenience store nearby the corner ... You drive through some ... low-income areas- you don’t see a green grocer but you see lots of convenience stores so those they’ll eat [those] foods. It’s not what’s ideal for their health. So that’s another thing I see. You can’t access food. The disparity is in ... accessing it because they don’t have transportation.*


The third barrier identified reflects the stigma towards asking for help with obtaining food. WIC staff discussed the WIC clients’ emotional experiences with requesting help and the impact that it can have on accessing appropriate services. One focus group participant depicted the stigma some mothers experience: 


*It’s very shameful to have to go somewhere ... and feel like you are begging for [food]. And if you get judged by somebody, then you really feel bad. And somebody ... might have had a bad situation before they came [to WIC where they] asked for food and [were] treated bad. So they’re very cautious about talking to somebody about getting food.*


WIC staff highlighted the need for nonjudgmental spaces where women in the perinatal period can feel safe enough to ask for and access the appropriate nutrition for themselves and their families. This theme underscores the challenges that WIC clients often face regarding food insecurity and food provision for themselves and their families. 

### 3.3. Barriers Preventing Perinatal WIC Clients from Attaining Food Security and Necessary Mental Health Services

The third theme reflects the challenges faced when seeking resources aimed at overcoming food insecurity and depression. The challenges identified by WIC staff include: (1) cultural factors such as language, immigration status, and lack of familiarity with local foods; (2) lack of adequate mental healthcare services; (3) lack of adequate supportive services such as financial support, safe and affordable housing, local food pantries, crisis nurseries, appropriate referral networks, and coordinated care within systems; (4) stigma; and (5) physical isolation such as living alone or with low support. 

The most consistent feedback from study participants is that WIC clients face many barriers to accessing both recommended foods (e.g., fruit, vegetables, whole grains) and adequate mental health services. Participants noted that, while there are some programs in place to provide services to low-income individuals:


*“...[n]ot every agency accepts the medical card or ... does ... free based counseling without pay or sliding fee scale.”*


Staff shared that many WIC-individuals could not afford the cost of therapy even with support from a sliding-scale system. Staff also highlighted that lack of coordinated care between mental health and medical providers significantly disrupts patient care and access to adequate treatment for women and pregnant people. Additionally, the lack of knowledge around available mental health resources or strategies for treating women and pregnant people with comorbid mental health concerns may exacerbate difficulties in coordinated care. As one participant stated: 


*We will call to try to make referrals for these women to our OBs … they will get denied access because the OBs don’t have knowledge of how to treat [women experiencing mental illness] and or they will start to treat them and they will call us and say we have got to discharge them because we just can’t handle their needs.*


Participants stressed the need for improving access to mental health services and coordinated care due to the rising number of mothers and pregnant people presenting to WIC providers with a myriad of mental health concerns.

Another common response among WIC staff addressed the stigma of being a person who experiences food insecurity. Study participants noted that some individuals experience a pressure to present in a certain way to avoid judgment or be seen as lesser-than because of their socioeconomic circumstances. One participant recalled an interaction they had with a WIC client, and shared: 


*“...she was saying … ‘you don’t want to look poor, you don’t want anyone to perceive you as poor’ … [so] a lot of the time their stuff is … a smoke screen.”*


Cultural factors such as language, familiarity with certain foods, and immigration status were also noted as potential barriers to care. For instance, participants shared that, for many, concern about becoming involved with immigration or other U.S. government entities can lead to considerable worry, and ultimately, prevent some individuals from asking for help. One participant recalled related experiences they had with WIC clients shared, “... when we start telling them … they can [get food delivered] if they’re not documented, then [they say] no … I don’t necessarily need [the food] because maybe they have [the] fear that somebody comes to [their] house.” Additionally, assumptions made around an individual’s level of familiarity with certain foods and their preparation, or around their fluency in reading the labels written in English, were identified as challenges that are often not considered. For example, one participant illustrated this point through an experience she had with a pregnant, WIC client who was diagnosed with gestational diabetes:


*… she’s opening her pantries asking me, can I eat this? Can I eat that? And it was like … American food. They gave her the ingredients to make tacos. She pulled the sour cream out [and asked] ‘can I eat this?’ [I said] [n]o you don’t eat that, you put it on things. But the assumption was made that she would know how to cook tacos with it.*


The discussion among study participants revealed several barriers to attaining food security and mental health treatment, including some that arise as consequences of assumptions made by supportive programming. Staff suggested that the barriers can result in perinatal WIC clients feel physically isolated from others, or as if they are on a “lonely island.” 

The overarching theme of the focus-group discussions centered on the complexity of the experiences faced by WIC clients. Staff noted that it is often difficult to focus on any one issue when these individuals so often face a multitude of high-priority obstacles. As summarized by one participant:


*[They will say] ‘I’m about to…be evicted from my apartment. There’s no food in the apartment. The power’s off. My refrigerator doesn’t work anyway because I don’t have any power …’ What takes precedence? So you know we can zero in on food. We can do something about that in most cases, but the other stuff is harder.*


## 4. Discussion

To our knowledge, this is one of the first studies to probe the collective experience of front-line WIC staff to identify the unique challenges faced by WIC clients. It is important to understand the complex nature of these clients’ experiences so that health care staff may identify intervention opportunities and subsequently improve systems-level processes when caring for this population. This qualitative study of food insecurity and perinatal depression described three key themes of issues that WIC clients face, as identified by WIC staff: (1) depression; (2) food insecurity; and (3) barriers to attaining food security and mental health resources. 

Addressing food insecurity and perinatal depression among women and families living in a low-income context is complex, with a potential bidirectional relationship between the two factors. Research examining the influence of food insecurity on mental health outcomes in pregnant and non-pregnant women is robust (see [13] for review); however, there is a growing body of literature that highlights the influence of depression on the development and maintenance of food insecurity. Garg, et al. [14] reported longitudinal associations between maternal depression and food insecurity such that mothers who endorsed depression at earlier time points were more likely to remain food-insecure over time compared to their non-depressed counterparts. It may be that the experience of depression in the perinatal period leads to difficulties with finding (or maintaining) employment or actively engaging with available resources, and ultimately, exacerbating the experience of food insecurity and/or depressive symptoms. These findings highlight the cyclical nature of this relationship and underscore the need for interventions aimed at disrupting these processes. 

Certain factors appear to amplify the influence of depression on food insecurity and vice versa. Research has focused on the identification of factors that increase the risk of perinatal depression symptoms, for example, Sidebottom, et al. [15] found that characteristics including maternal age, having experienced abuse, not living with the infant’s father, lack of social support, lack of phone access, or experiencing food insecurity were all associated with reported depression during the perinatal period. The authors suggested that on-going screening across this period was needed in order to accurately capture the full range of reported symptoms. Low-income women have also described difficulty obtaining basic needs such as housing, employment, and transportation and these factors may significantly increase stress and the experience of depressive symptoms [16]. In the present study, WIC staff echoed the sentiment that stress and mental health concerns such as perinatal depression are not isolated occurrences, but rather can be triggered or amplified by a lack of resources and barriers to care. Given that women living in a low-income context are especially vulnerable to significant and variable daily stressors, ongoing monitoring, screening, and assessment of needs across the perinatal period is imperative for adequate and timely interventions.

One efficient intervention that should be normalized for perinatal women is a basic food security assessment along with a mental health assessment. In a small pilot study, Browne and Ponce [17] found that most community mental health providers do not ask about food insecurity, even when they were provided with training on this topic. Others have reported that using a brief food security instrument can accurately identify people requiring interventions or referrals. For example, Gundersen and colleagues [18] found that asking patients whether they were, (1) worried they food would run out before they could afford to buy more, and (2) how often the food they bought did not last before they had the means to purchase more, accurately identified household food insecurity. WIC staff in our study agreed that food insecurity of WIC clients was not adequately assessed, and this may be due to limited knowledge of food insecurity, of available resources, or of perceived precedence of other problems client may be facing. Future research should examine the most effective method of assessing food insecurity to elicit accurate information from participants while minimizing shame or experience of stigma. 

Healthcare providers are well positioned to address commonly co-occurring mental health and food needs in low-income populations. Previous work has highlighted the positive impact of coordinated care within primary care clinics on those experiencing serious mental health and medical concerns [19,20,21]. One study found coordinating low income pediatric medical care with food as prescription through a partnership with a local farmer’s market yielded overarching positive results in improving access to fresh or nutritious food [22]. Given these promising outcomes, it is reasonable to suggest that WIC clinics work to increase implementation of coordinated care in an effort to more efficiently address the complex and highly impactful needs of low-income mothers and their families. Future studies should examine other methods of coordinated care among education, public, healthcare, and private sectors to best treat food insecurity and peripartum mood disorders concurrently.

Despite the strengths of this qualitative study with a sample WIC staff from three nonmetropolitan counties, our study is not without limitations. First, our study is limited in scope due to the in-depth nature of qualitative investigations and our sample is limited to a small range of perspectives. Accordingly, this study presents the potential for replication with clinic staff in other regions of the country to identify if the perceptions of food insecurity and perinatal depression are consistent. Second, addressing food insecurity and other social needs in health care settings presents a conflict for some types of providers and compliance issues when offering good and services to encourage individuals to use some mental health services. For this study, we sought to identify how providers perceive the relationship between food insecurity and perinatal depression and learn which actions they take. Accordingly, not soliciting providers views on conflict of interests and compliance issues of extending social provisions to receive mental health care represent a minor limitation for this study given that the staff worked in social service settings. 

## 5. Conclusions

This qualitative focus group study examined WIC staff’s perceptions of the needs of their clients and the association between food insecurity and depression among some staff in the Midwestern United States. Three themes were identified: (1) Depression Experienced by Perinatal WIC Clients, (2) Food Insecurity in Perinatal WIC Clients, and (3) Barriers Preventing Perinatal WIC Clients from Attaining Food Security and Necessary Mental Health Services. Study participants highlighted the considerable obstacles experienced by low-income, perinatal women in their attempts to access quality services. The complex nature of the relationships between depressive symptoms and food insecurity in the perinatal period suggests that efforts to implement robust coordinated networks within WIC programs would be a promising step toward creating seamless care for this population. Future research aimed at assessing feasibility of implementation of coordinated care efforts in WIC centers is needed. 

## Figures and Tables

**Figure 1 healthcare-11-00068-f001:**
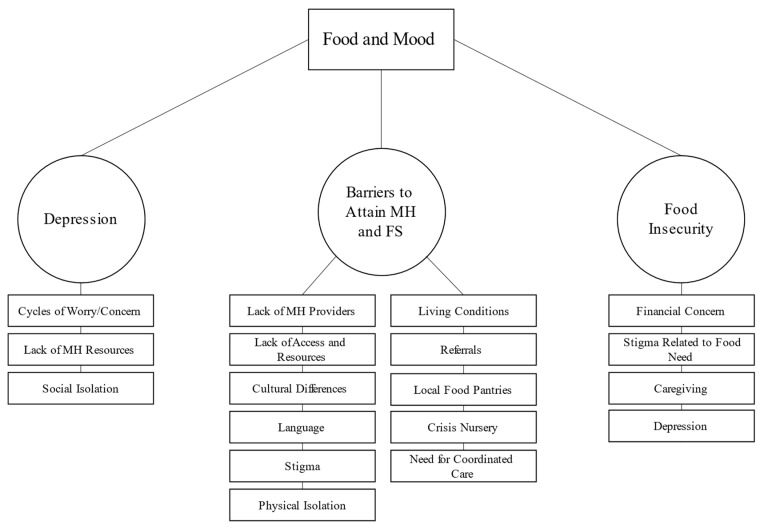
Thematic Network Analysis of Results from Interviews with Staff. MH = mental health; FS = food security.

## Data Availability

The data presented in this study are available on request from the corresponding author. The data are not publicly available due to privacy concerns.

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
