# Peer review of "WIC Staff Views and Perceptions on the Relationship between Food Insecurity and Perinatal Depression"

_healthcare, 2022, doi:10.3390/healthcare11010068_

Round 1

Reviewer 1 Report

The article presents a qualitative study based on focus groups examining WIC  staff’s perceptions of the needs of 361 their clients and the association between food insecurity and depression. The study found interesting results which can be useful for both research and practice. I do not have particular concerns with the study and I think the overall merit of the paper is high which makes me feel that the paper can be published as it is. Perhaps, the authors may offer better indications about the realization of robust networks toward creating seamless care for this population (as they say in the conclusion). I would not recommend you add an additional paragraph in the conclusion but maybe you explicit more about how you would realize such implications. This could be useful for practitioners or launch project realization.

Author Response

The article presents a qualitative study based on focus groups examining WIC staff’s perceptions of the needs of 361 their clients and the association between food insecurity and depression. The study found interesting results which can be useful for both research and practice. I do not have particular concerns with the study and I think the overall merit of the paper is high which makes me feel that the paper can be published as it is.

Response: Kindly received feedback. Many thanks for this.

Perhaps, the authors may offer better indications about the realization of robust networks toward creating seamless care for this population (as they say in the conclusion). I would not recommend you add an additional paragraph in the conclusion but maybe you explicit more about how you would realize such implications. This could be useful for practitioners or launch project realization.

Response: Thank you for this feedback. We reduced the emphasis of our original conclusion. We also added a limitations section to help readers see our limits and possible challenges to realization.

Reviewer 2 Report

This study is the first to analyze the perception of staff from Special Supplemental Nutrition Program for Women, Infants and Children (WIC) on the relationship between food insecurity and perinatal depression within the clients from WIC. The paper is well written and the theme is highly important and concerning. 

However, the limited number of participants is too low (24) to extrapolate and conclude that significant changes (policy changes as mentioned in the text) are needed. Also, this paper presents most likely preliminary results and not conclusive ones. Other themes could emerge as well from more larger group interviews. A limitations section is needed (i.e., perception of WIC staff can be heavily biased and misinterpreted).

Since the authors have data and resources (i.e., SPSS) a more comprehensive image would be delivered if the authors perform statistical analysis perhaps to show relationships such as correlations or regressions (predictors) between the identified themes and the factors that are influencing them; i.e., what influences depression, how much of “Depression” is explained by the mentioned factors: cycles of worry/concern, lack of MH resources, and social isolation (variance %). This way a clearer image would be obtained on the relationship between food insecurity and perinatal depression.

Author Response

This study is the first to analyze the perception of staff from Special Supplemental Nutrition Program for Women, Infants and Children (WIC) on the relationship between food insecurity and perinatal depression within the clients from WIC. The paper is well written and the theme is highly important and concerning. 

Response: Kindly received feedback. Many thanks for this.

However, the limited number of participants is too low (24) to extrapolate and conclude that significant changes (policy changes as mentioned in the text) are needed. Also, this paper presents most likely preliminary results and not conclusive ones. Other themes could emerge as well from more larger group interviews. A limitations section is needed (i.e., perception of WIC staff can be heavily biased and misinterpreted).

Response: Thank you for this feedback. We have worked to clarify the draft throughout. We have also added a limitations section as follows:

Despite the strengths of this qualitative study with a sample WIC staff from three nonmetropolitan counties, our study is not without limitations. First, our study is limited in scope due to the in-depth nature of qualitative investigations and our sample is limited to a small range of perspectives. Accordingly, this study presents the potential for replication with clinic staff in other regions of the country to identify if the perceptions of food insecurity and perinatal depression are consistent. Second, addressing food insecurity and other social needs in health care settings presents a conflict for some types of providers and compliance issues when offering good and services to encourage individuals to use some mental health services. For this study, we sought to identify how providers perceive the relationship between food insecurity and perinatal depression and learn which actions they take. Accordingly, not soliciting providers views on conflict of interests and compliance issues of extending social provisions to receive mental health care represent a minor limitation for this study given that the staff worked in social service settings.

Since the authors have data and resources (i.e., SPSS) a more comprehensive image would be delivered if the authors perform statistical analysis perhaps to show relationships such as correlations or regressions (predictors) between the identified themes and the factors that are influencing them; i.e., what influences depression, how much of “Depression” is explained by the mentioned factors: cycles of worry/concern, lack of MH resources, and social isolation (variance %). This way a clearer image would be obtained on the relationship between food insecurity and perinatal depression.

Response: Dear Reviewer thank you for this feedback. You do raise a good point. Our research study was qualitative, so rather than conducting any sort of hypothesis testing we sought to generate a broad report of what frontline workers are thinking. The sample of 24 is far too small for quantitative work (as you know) and we would not have sufficient power for an n:p ratio. The sample of 24 for a qualitative study is robust (as you know) as should have major impact for WIC programs nationally. We also plan to disseminate this work widely with a discussion guide to help foster initial discussions with interested parties. Thank you again for your constructive review.

Round 2

Reviewer 2 Report

The paper can now be accepted for publication.